# Stress and Bio-Ethical Issues Perceived by Romanian Healthcare Practitioners in the COVID-19 Era

**DOI:** 10.3390/ijerph182312749

**Published:** 2021-12-03

**Authors:** Daniela Reisz, Iulia Crișan, Andrea Reisz, Raluca Tudor, Doina Georgescu

**Affiliations:** 1Department of Neurology, University of Medicine and Pharmacy “Victor Babeș”, 300041 Timișoara, Romania; reisz_daniela@yahoo.com (D.R.); tudor.raluca@yahoo.com (R.T.); 2Department of Psychology, West University of Timișoara, 300223 Timișoara, Romania; 3Department of Letters, History, and Theology, West University of Timișoara, 300223 Timișoara, Romania; andrea_reisz@hotmail.com; 4Department of Internal Medicine, University of Medicine and Pharmacy “Victor Babeș”, 300041 Timișoara, Romania

**Keywords:** COVID-19 pandemic, healthcare practitioners, bio-ethical principles, stress, wellbeing, decision making, autonomy, challenges

## Abstract

Objective: The COVID-19 pandemic had a major impact on different areas of life, especially in the medical system. Because of the pandemic outbreak, the medical system was remodeled to treat COVID-19 patients in secure conditions. Those changes and restrictive measures have put pressure on individual adaptability. The present study investigated the stress of Romanian healthcare practitioners (HCP) and the capacity to deal with new bio-ethical issues that emerged during the COVID-19 pandemic in 2020. Methods: We analyzed results from a survey on 97 Romanian HCP in several areas: personal experience with COVID-19, perceived emotional distress, and appraisal of bio-ethical principles respected or infringed during the pandemic in 2020. Results: Unlike previous studies, our respondents reported low to moderate stress levels. In addition, few bio-ethical principles were infringed on a personal level. Tendencies to sacrifice individual autonomy and make decisions affecting patients and co-workers were more prevalent among HCP with over 30 years of experience. Conclusions: Retrospectively, Romanian HCP in our sample appeared to share an embellished view of the COVID-19 pandemic in 2020. Potentially related factors and coping mechanisms with stress are discussed.

## 1. Introduction

The COVID-19 pandemic is a rapidly evolving situation with an unpredictable course of development and unexpected outcomes. Starting with the first wave in 2020, associated with a lack of information and uncertainty regarding the virus and its impact on the human organism [1], up to the fourth wave in autumn 2021, the worldwide pandemic has been continuously generating numerous shifts and changes on multiple levels. The medical systems of the affected countries were faced with medical, legal, and ethical challenges [2], which were reflected on a social and individual level. As stated by numerous authors and currently experienced by nations worldwide, the pandemic has inflicted a crisis that caused a shift from the individualistic perspective on personal wellbeing and ethics to a collectivistic conception of social health and safety [3].

Since the pandemic outbreak, numerous theoretical and empirical papers have been published reflecting a growing global concern with managing this crisis and dealing with social and healthcare implications. Several topics in this direction include articles on health inequities that emerged during the COVID-19 pandemic [3,4,5], ethical considerations and recommendations for hospital personnel [1,2,6,7], and research on the psychological impact of the pandemic on different populations [8,9,10]. In this regard, the present study aims at exploring several aspects related to the appraisal of COVID-19 by Romanian medical professionals in the year of the pandemic outbreak.

### 1.1. Stress and Mental Health Related to COVID-19

A rapidly growing body of literature has been investigating the psychological impact of the pandemic on different samples since 2020. There is great heterogeneity associated with the empirical results, dependent on, among others, the type of study (e.g., national or international) and the targeted population (e.g., general population, healthcare professionals, etc.). For instance, an international study on the effect of national lockdowns on the general population’s mental health in 2020 revealed that 50% of respondents from 78 countries reported moderate levels of mental health and identified social support, psychological flexibility, and educational level as significant predictors of mental health [8]. Consistent with such results, a cross-sectional survey on Hong Kong citizens revealed that psychosocial flexibility and a prosocial attitude mediated the relation between mental health and perception of COVID-19 [10].

On the other hand, empirical studies on healthcare practitioners (HCP) often paint a different picture since this professional category has been placed in the front line of combat against the pandemic. A national study investigating stress-related variables in HCP in Cyprus found high levels of depression and PTSD in about 15% of the sample, with nurses being more prone to these symptoms than the other categories, including doctors [11]. In addition, a multinational study on HCP identified symptoms of burnout in 67% of participants, which were significantly predicted by patient-facing roles (e.g., doctors or nurses), professional redeployment, anxiety, depression, safety attitudes, and being tested for SARS-CoV-2 [12]. Other studies [13,14] found high levels of depression, anxiety, and sleep disorders in HCP with high exposure to COVID-19-positive patients. A qualitative study exploring HCP experiences of awaiting COVID-19 test results in Denmark revealed a solid professional identity consistent with abnegation and dedication to their role despite the perceived risk of being infected and threat to their health [15]. A study on Chinese HCP revealed infection of co-workers and family members, scarcity or unfair distribution of protective equipment, and ethical issues as the most frequent concerns [16].

This framework raises the need for empirical contributions to illustrate healthcare professionals’ attitudes towards the pandemic and the associated stress and its effects. As the pandemic affects nations worldwide, a cross-cultural perspective on such variables is needed. While research on the stress of Romanian HCP in the pre-pandemic period has been scarce, interest in this topic has increased exponentially since 2020. Previous studies investigating mental health issues in Romanian HCP related to the outbreak of the COVID-19 pandemic yielded heterogeneous results. Some studies revealed moderate levels of fear of contracting the disease or moderate levels of stress and high levels of anxiety in the first semester of the pandemic [17,18]. Another study reported high levels of burnout in medical residents (i.e., with 0–5 years of experience), questioning the adaptability of less experienced HCP to the pandemic compared to their senior colleagues [19]. However, to our knowledge, no study has compared Romanian HCP with different professional experience regarding their appraisal of COVID-19 related stress. To this end, the present study addresses the investigation of such differences.

### 1.2. The Four Bio-Ethical Principles

For the present study, we took into account the four central bio-ethical principles stated by Beauchamp and Childress [3]: beneficence (the imperative of doing good), nonmaleficence (the imperative of not doing harm), autonomy (norms to respect autonomous decisions), and justice (norms regarding the fair distribution of costs, risks, and benefits). Consistent with Page’s conceptualization [20], we included truth (the obligation to transmit accurate information regarding the virus, the disease, epidemiology, prognosis, medical procedures, treatment, etc.) as a fifth separate principle.

Although the original authors argued against the prioritization of one principle over another, other studies have revealed that this vision has shifted over time. For instance, pre-pandemic studies revealed nonmaleficence as the most important principle among psychology students and social workers [20,21]. While Landau and Osmo [21] did not investigate gender, Page’s study found no gender differences in ranking the principles except for truth-telling, which women valued more than men [20].

Following the COVID-19 pandemic outbreak, several papers have stressed a practical reconsideration of the principles. For instance, Brenna and Das [3] argued that during the pandemic, justice becomes unique among the four principles, as it creates tension within the principle. The tension is reflected as distributive justice superseding individual justice (i.e., finding the optimal solution for individual patients), a central principle in clinical practice that guides the clinician–patient relationship [22]. This view is shared by authors recommending the constant allocation of medical efforts towards fair resource distribution [23].

In their recent meta-analysis on the bio-ethical implications of the COVID-19 pandemic in low-prevalence countries, Skapetis and colleagues [24] signaled a lack of information regarding common ethical issues encountered in such countries. In addition, studies investigating the attitudes towards ethical principles prior to the COVID-19 pandemic enrolled undergraduates [20] or social workers [21] as participants or evaluated ethical principles through attitudes towards specific fictional scenarios [25,26]. Page’s [20] results were consistent with Landau and Osmo’s [21] and showed no correlations between interpreting a scenario and the practical use of principles. Therefore, a more practical framework for studying adherence to these principles is needed. Systematic reviews on bioethical implications for COVID-19 research have been conducted [24], yet knowledge regarding the impact of the pandemic on practicing these principles in healthcare systems is scarce. In addition, the lack of studies in Romania on the perception of ethical values by medical practitioners amidst the COVID-19 pandemic constitutes another reason for conducting the present study.

Considering this framework, we set the following research objectives:(1)To explore the subjective experience of stress and the bio-ethical challenges of Romanian healthcare practitioners during the COVID-19 pandemic in 2020.(2)To investigate differences between categories of healthcare workers in experiencing stress and bio-ethical challenges.(3)To assess a possible pattern of handling stress and bio-ethical challenges among HCP.

## 2. Materials and Methods

We constructed a cross-sectional survey with 18 items using SurveyMonkey and administered it to healthcare workers (HCP) from 3 July 2021 to 5 September 2021. HCP were recruited by convenience and snowball sampling. The questionnaire was distributed via social media (i.e., WhatsApp, Facebook) and by e-mail. The anonymity of participants was guaranteed, and no identification details were requested, including age and gender information. Gender was intentionally not included as a variable since we did not investigate gender-related differences impacting bio-ethical decision making. The questionnaire addressed all employee categories of public and private healthcare facilities, including physicians, nurses, psychologists, managers, physical therapists, pharmacists, technicians, and researchers who were active during the COVID-19 pandemic in 2020. The survey investigated the attitudes and practices of professionals during the COVID-19 pandemic in 2020. The authors generated a pool of items related to these issues. The research team evaluated the item pool and extracted the relevant items for the present objectives and context. These items were organized into five sections: demographics (4 items), personal experience with COVID-19 (2 items), level of stress and wellbeing throughout 2020 (2 items), attitudes and beliefs towards respecting or infringing ethical principles (8 items), and appraisal of mass-media information and the measures taken by the Romanian government (2 items). Types of questions included multiple-choice formats, rating scale questions, Likert scale questions, ranking questions, and open-ended questions. The English translation of the questionnaire is available in Appendix A. For the objectives of the present study, only the first four sections were analyzed.

### Data Analysis

We used SPSS for Windows to analyze the frequencies of participants’ responses for multiple-choice questions, which were computed as categorical variables. Means and standard deviations were computed for continuous variables. We ran normality tests to check the variables’ distribution and used nonparametric tests to investigate differences between respondent categories. Bivariate correlations and linear regressions were conducted to show relations between variables.

## 3. Results

A total of 97 respondents completed the survey. This sample represents about 0.04% of the general HCP population; according to a study using data provided by the National Statistics Institute, in 2019, Romania had 63,300 doctors, 18,100 pharmacists, 150,300 nurses, and 2200 physical therapists. With respect to the total number of citizens, Romania had one doctor per 307 inhabitants [27].

### 3.1. Demographics

The demographic characteristics of our sample are displayed in Table 1. Most respondents were from the western part of Romania (i.e., Banat—81.4%). The majority of respondents (68%) were doctors, and 27.8% had 20–30 years of experience. Inpatient facilities or hospitals were chosen as the primary activity sector by 50 respondents (51.5%), followed by outpatient facilities indicated by 25 participants (25.8%). We used the Kolmogorov–Smirnov and the Shapiro–Wilk tests of normality to check the distribution of demographics in our sample (i.e., region, occupation, years of professional experience). The results of both tests were statistically significant, indicating that our sample was not normally distributed (D (97) = 0.192–0.482, *p* = 0.000; W (97) = 0.468–0.880, *p* = 0.000).

### 3.2. Personal Experience with COVID-19

Items investigating respondents’ personal experience with COVID-19 had multiple-choice formats so that respondents were allowed to check all the possibilities that applied to them. Table 2 illustrates percentages of cases reflecting respondents’ personal experience with COVID-19; half of the sample (50.5%) reported having attended COVID-19-positive patients, and almost half (41.2%) had at least one family member or friend with the disease. Less than half of the respondents (37.1%) reported having acquired information about COVID-19 through reading. Of note, a quarter of the sample (24.9%) had experienced the disease as patients, with either mild, moderate, or severe symptoms. Concerning decision making during the pandemic in 2020, most respondents (62.9%) reported having been in the situation of deciding for family members, and 53.6% admitted being in the situation of making decisions that affected patients.

Next, we wanted to see whether there were differences between the levels of professional experience regarding the types of decisions affecting others (see Table 3).

Results showed significant differences between categories of professional experience regarding decisions that affected patients (χ^2^ = 15.651, *p* = 0.004) and colleagues (χ^2^ = 10.844, *p* = 0.028). Post hoc Mann–Whitney U tests revealed that healthcare workers with the least experience (0–5 years; mean rank = 31.82) made significantly fewer decisions affecting patients than each of the other categories, which were 5–10 years (U = 66, Z = −2.571, *p* = 0.01), 10–20 years (U = 121, Z = −3.31, *p* = 0.001), 20–30 years (U = 186, Z = −2.643, *p* = 0.008), and professionals with over 30 years of practice (U = 74, Z = −3.309, *p* = 0.004). As expected, HCP with over 30 years of experience made more decisions affecting co-workers than two other categories, 0–5 (U = 92, Z = 2.935, *p* = 0.003) and 10–20 (U = 107, Z = 2.212, *p* = 0.027).

Correlations between variables describing personal experience with COVID-19 and variables reflecting decisions affecting others were computed (see Table 4). Treating COVID-19-positive patients correlated positively with making decisions that affected patients (*r* = 0.526, *p* = 0.001) and negatively with making no decisions affecting others (r = −0.323, *p* = 0.001). Having COVID-19-positive family members correlated positively with making decisions affecting family members (*r* = 0.253, *p* = 0.012) and colleagues (*r* = 0.225, *p* = 0.027). Reading about the virus and vaccine showed positive relations with decision making affecting family members (*r* = 0.237, *p* = 0.02). On the other hand, having no direct experience with COVID-19 was negatively correlated to making decisions affecting patients (r = −0.288, *p* = 0.004). Interestingly, suffering the loss of a relative or close friend correlated positively with making decisions affecting colleagues (*r* = 0.246, *p* = 0.015). No significant covariations were found between having personally experienced COVID-19 in either form (i.e., with mild or moderate–severe symptoms) and making decisions affecting others (*r*s > 0.172, *p*s > 0.05).

### 3.3. Psychological Impact of the Pandemic

Descriptive statistics for variables reflecting the psychological impact of the pandemic on our respondents are presented in Table 5. General wellbeing throughout 2020 was appraised as moderate (m = 2.93, SD = 1.15) by 41.2% of participants. Among the stress-related variables, concern showed the highest central tendency (m = 2.23, SD = 0.823), pointing to an average moderate level in our sample.

Frequencies were then computed to show the percentage of a particular response (see Table 6). The participants’ general wellbeing was rated as moderate by 41.2% of respondents, with 21.6% reporting a high level and 16.4% a very low level of wellbeing. Results regarding the affective symptomatology most frequently endorsed by participants yielded moderate irritability (51.61%) and anxiety (37.63%), as well as moderate (41.94%) and high (32.26%) concern. Approximately one-third of participants reported seldom relaxation difficulties (36.56%) and anticipation of a negative event (39.78%), while 29% reported having relaxation difficulties most of the time. Kruskal–Wallis tests revealed no differences between categories of professional experience regarding the appraisal of wellbeing and stress (χ2 (4) = 0.372–6.017, *p* > 0.05).

Then, we wanted to check the degree to which stress-related variables (Model 1) and personal experience with COVID-19 (Model 2) predicted general wellbeing throughout 2020 (see Table 7). A significant regression equation was found for Model 1 (F (5, 91) = 5.326, *p* = 0.001), with an R2 of 0.226. Of the stress-related variables, only the level of irritability was a significant negative predictor of the general wellbeing throughout 2020. On the other hand, no significant regression equations were found for Model 2 (F (8, 88) = 0.787, *p* = 0.616, R2 = 0.067), indicating no significant relation between the personal experience with COVID-19 and the general wellbeing of our sample throughout 2020.

### 3.4. Values and Bio-Ethical Principles

Table 8 presents a synopsis of participants’ reactions towards the infringement of ethical principles from a social (i.e., which principles they thought were violated due to the official restrictive measures) and a personal perspective (i.e., which principles they had to infringe). Almost half of the sample (47.4%) rated justice as the principle primarily violated on a social level. Interestingly, more than half of the sample (58.8%) declared they had not violated any of the above-mentioned principles on a personal level. One-third of the sample (33%) appraised beneficence as the hardest principle to infringe. By comparison, autonomy was rated as personally the most infringed principle and the easiest to infringe after justice.

We computed Kruskal–Wallis tests between categories of medical experience to check for differences in the appraisal of principle infringement. Results revealed no significant differences for most principles, except for autonomy on a personal level (see Table 9). Post hoc Mann–Whitney U tests showed that professionals with over 30 years of experience reported having personally broken this principle significantly more than other categories, 0–5 (U = 92, Z = 2.935, *p* = 0.003) and 20–30 (U = 132, Z = 2.311, *p* = 0.021).

We computed multiple logistic regression analyses to see whether stress-related variables predicted the appraisal of principle infringement on both the social and the personal level. On a social level, anxiety (B = 0.848, SE = 0.374, Wald = 5.137, *p* = 0.023) was found to significantly positively predict the infringement of justice [OR = 2.336, 95% CI (1.122, 4.865)]; concern (B = 0.993, SE = 0.387, Wald = 6.597, *p* = 0.001) positively predicted the infringement of truth (OR = 2.701, 95% CI (1.265, 5.763)). On a personal level, irritability (B = 1.358, SE = 0.668, Wald = 4.127, *p* = 0.042) positively predicted the infringement of beneficence (OR = 3.887, 95% CI (1.049, 14.403)), and relaxation difficulties (B = −1.343, SE = 0.654, Wald = 4.213, *p* = 0.04) negatively predicted the infringement of truth (OR = 0.261, 95% CI (0.072, 0.941)).

The last item of the survey requested participants to assess situations where bio-ethical principles were questioned on a personal level from a list of 20 possible situations (see Table 10). The most frequently endorsed situations were limiting the access of families to the hospital (51.5%), neglecting to assist chronic patients (41.1%), and the “blind” application of protocols (32.36%). In addition, 31.18% of the sample reported having to decide between personal safety and carrying out professional duties.

To check for differences in encountering these situations, we ran comparisons on two levels: categories of professional experience and HCP who attended vs. HCP who did not attend COVID-19 patients (see Table 10). Results of Kruskal–Wallis tests revealed marginal differences between HCP with different experience concerning two situations: neglecting the assistance of chronic patients (χ2 = 9.166, *p* = 0.057) and managing hospital sections at surge capacity (χ2 = 9.401, *p* = 0.052). Post hoc Mann–Whitney U tests showed that professionals with less than 5 years of experience encountered the first situation (i.e., neglecting chronic patients) significantly less than three other categories: 10–20 (U = 165, Z = −2.287, *p* = 0.022), 20–30 (U = 194.5, Z = −2.536, *p* = 0.011), and professionals with over 30 years of experience (U = 99.5, Z = −2.558, *p* = 0.011). Differences between groups related to managing sections at surge capacity were more heterogeneous; HCP with 5–10 experience years encountered this situation more than HCP with 0–5 years (U = 82.5, Z = 1.98, *p* = 0.048), 10–20 years (U = 82.5, Z = 1.98, *p* = 0.048), and HCP with over 30 years of experience (U = 50.5, Z = 2.274, *p* = 0.023). Interestingly, HCP with over 30 years of experience appeared to have encountered the situation less than HCP with 20–30 years of experience (U = 148.5, Z = −1.921, *p* = 0.055). On the other hand, Mann–Whitney U tests revealed 13 situations that HCP treating COVID-19 patients experienced significantly more than HCP who had no experience treating COVID-19 patients.

## 4. Discussion

The present study explored the impact of the COVID-19 pandemic on the stress perceived by Romanian medical professionals and the bio-ethical challenges they faced throughout 2020. We intended to capture a national perspective by including practitioners all over the country in the survey. However, the fact that most responses came from professionals in the western part reflects a regional perspective on the pandemic rather than a national point of view, as perspectives might differ from one part of Romania to another. In addition, the majority of responses (68%) came from doctors, which focuses our investigation on this category of HCP. Concerning the distribution of professional experience, three of the five categories included similar proportions of respondents (22.7–27.8%), with professionals with the least experience (0–5 years) being the least represented category (11.3%). The distribution of practitioners across activity sectors showed that 51.54% of the sample were active in hospitals, and 25.77% worked in outpatient facilities, pointing to a direct experience of most respondents with COVID-19 patients during 2020.

### 4.1. Personal Experience with COVID-19

The analysis of variables related to personal experience and decision making impacting others revealed that half of the sample attended COVID-19-positive patients, and 53.6% reported having been in the situation of making decisions affecting patients. The positive correlation between these variables indicates a direct relation between assuming and exerting a professional healthcare role and deciding for the patients under care. In addition, treating COVID-19-positive patients correlated negatively with making no decisions affecting others, suggesting a common variance for attending patients and making decisions affecting others. This finding may indicate an awareness of moral responsibility regarding others (we suppose that every adult person was in the position of deciding for others in 2020). Consistent with previous studies [11,15], our results underline the professional identity of HCP in times of the pandemic. Still, unlike prior research, this role appears to be directly linked to assuming control over the patients’ liberty to decide. In addition, other vicarious experiences, such as having COVID-19-positive relatives or suffering the death of a relative or friend, showed a direct relation to deciding for family members and colleagues, as did acquiring information about the disease. Such attitudes may be viewed as gnawing at the bio-ethical principle of autonomy in an extended ethical framework.

However, no relations were found between personally going through the disease in either form (i.e., asymptomatic–mild or moderate–severe) and making decisions affecting others. This finding may indicate a certain detachment from their personal experience in assessing situations involving others. It may also point to a lack of insight regarding the personal power to decide and its possible consequences. On the other hand, a small percentage of the sample (5.2%) denied having been in the position to decide for others, which might be less probable in the medical context during the 2020 development of the pandemic. Such implications raise questions regarding HCP understanding of making decisions affecting others, which on a conceptual level might imply a certain level of ethical illiteracy.

HCP with different professional experience lengths demonstrated significant differences in making decisions affecting patients and colleagues. Specifically, a higher tendency towards these types of decision making was found in professionals with more years of experience (e.g., 20–30, above 30) than in HCP with less experience (e.g., 0–5, 5–10 years). Such findings were expectable since, at least in Romania, superior positions in hierarchical structures of the healthcare system are usually occupied by doctors with long-term professional experience. Thus, this type of position might enable them to make decisions affecting others in times of crisis, such as a rapidly evolving pandemic. In addition, experienced doctors usually have more patients under their care than less experienced practitioners, which might increase the frequency of deciding for some of these patients.

### 4.2. The Psychological Impact of the Pandemic

Unlike studies reporting high levels of burnout or stress-related symptoms in HCP [11,12,13,14,16,19], the results of our sample pointed to moderate general wellbeing throughout 2020, with seldom relaxation difficulties and negative anticipation. Nevertheless, 27% of the sample reported frequent relaxation difficulties. In addition, except for irritability, general wellbeing was not significantly predicted by stress-related variables. In a year that saw the implementation of socially restrictive measures (e.g., lockdowns, traveling limitations, etc.), such findings may indicate a certain unawareness of such clinical signs of stress or a diminished insight into or dissociation from the subjective experience of stress. In this regard, finding no relation between general wellbeing and personal experience with COVID-19 may support such interpretations. On the other hand, in 2020 in Romania, the medical personnel were privileged by fewer restrictions during lockdown (e.g., allowing mobility), which might have contributed to lower levels of perceived stress. We note that Romania’s pandemic peak in 2020 lasted for about two months and did not overwhelm the medical system.

Compared to the results of other studies on Romanian HCP conducted at the beginning of the pandemic [17,18], our sample appeared to overrate the general wellbeing and underrate the level of stress. Such responses may indicate a way of coping with extreme stress in the form of negation (parallel with coping with trauma—forgetting traumatic events). On the other hand, results may also suggest an ability to adapt to stressful situations; this being a retrospective study, respondents might have used effective healthy strategies to cope with stress given their medical experience. In line with other studies on other cultures [15], results could point to the tendency in our sample to compensate through a sense of usefulness in the work field, fulfilling their purpose as healthcare workers, or assuming healer roles.

### 4.3. Values and Bio-Ethical Principles

In line with the aforementioned tendency to make decisions affecting others, HCP with the most extensive experience reported having infringed autonomy on a personal level more frequently than less experienced HCP. In addition, this principle, alongside justice, was generally rated as the least hard to infringe principles. From a cultural standpoint, partially due to its Communist heritage, Romania relies not on respecting individual liberty but on a patriarchal tradition that values family relations and obedience to authority. In this framework, the authority attributed to doctors with high educational statuses and their long professional experience in treating patients might contribute to the belief that patients must abide by the HCP’s indications. Both HCP and patients often share this autonomy-discounting perspective. On the other hand, due to the popularization of democratic principles in the post-Communist decades and their inclusion in the official educational curricula, younger Romanian HCP might have gained more awareness regarding the infringement of such principles.

However, a certain level of ethical illiteracy is not to be discounted, especially since our results revealed an overly positive self-image shared by approximately 60% of the sample, reported as not having broken any bio-ethical principles on a personal level. Concerning the hardest principles to infringe, HCP did not stray away from respecting beneficence and nonmaleficence. These findings are consistent with previous studies on hierarchies of ethical principles among different populations [20,21]. Autonomy and justice were rated last in the hierarchy, supporting previous indicators of our sample’s tendency to limit individual freedom.

Along with autonomy, justice and truth were rated as the most frequently infringed principles on a social level. In the Romanian society of 2020, justice may refer to several aspects, such as the distribution of resources, neglecting chronic patients, or re-organizing the medical system to support the surge of patients. For instance, consistent with articles highlighting these issues [3,16,23], the Romanian medical system has seen several hospital sections (e.g., orthopedics, dermatology, cardiology) closed or re-organized into sections for COVID-19 patients. Such measures have definitely impacted Romanian HCP in the first year of the pandemic. Another factor that might have influenced respondents’ appraisal of the infringement of truth could be the conflicting information on the effectiveness of several drugs used to treat infections with COVID-19 (e.g., Plaquenil, Dexamethasone, Azithromycin) that circulated in the medical world in 2020. Such information might have decreased the trust in the integrity of scientific information, which in return reflected upon the appraisal of truth.

On the other hand, truth was rated the third hardest principle to infringe after beneficence and nonmaleficence, reflecting a preoccupation of our sample with respecting this principle. Nevertheless, personally infringing the truth was associated with decreased relaxation difficulties, pointing to our sample’s ambivalence regarding this principle. Of note, as prior research has revealed a higher valorization of this principle in women than in men [20], responses in our sample may be biased by unequal gender proportions, which we did not investigate.

Several other stress-related variables were positively linked to principles’ infringement, indicating a relation between appraised social dynamics and subjective distress; our sample appeared to witness the social infringement of justice with increased anxiety and that of truth with growing concern. On a personal level, increased irritability was linked to infringing beneficence (i.e., the more irritable, the greater the probability to disrespect beneficence), but only for a small percentage of the sample.

The most frequently reported bio-ethical issues were consistent with previous articles and were encountered to a higher degree by HCP with direct experience in treating COVID-19 patients. They highlighted the neglect of chronic patients [23], the limitation of family access to hospitals [1], managing inpatient sections at surge capacity [3], and difficulties in handling personal safety while carrying out professional duties [7,12]. However, while recommendations to systematically follow protocols have been issued since the pandemic outbreak [2,6], the “blind” application of protocols was one of the most frequently encountered situations reported by HCP with and without experience in attending COVID-19 patients. Differences between categories of professional experience were found for neglecting chronic patients, a situation less encountered by less experienced professionals. However, results revealed that professionals with over 30 years of experience had to manage hospital sections at surge capacity significantly less than less experienced categories (e.g., 5–10, 20–30). Such results contrast with this category’s elevated tendency to make decisions affecting others (e.g., patients or colleagues), which we linked to their superior positions in the healthcare system, motivated by a high professional status. The probability of delegating such responsibilities to subordinates who often are less experienced professionals is a factor that could account for such findings. In this regard, our results are consistent with previous national studies reporting higher levels of burnout in younger doctors working in regular hospital sections than in their senior colleagues [19]. On the other hand, the lack of significant differences between experience categories in perceiving stress throughout 2020 eliminates stress as a possible factor.

### 4.4. Limitations

The present study has several limitations that should be mentioned. First, the fact that most respondents were doctors limits the representativeness of the present sample to all categories of healthcare workers. In addition, the reduced sample size (0.1% of the general Romanian HCP population) and the fact that most participants were from the western part of Romania restrains the generalizability of the findings to the general Romanian HCP population. The sampling methods could have led to most respondents being recruited from the researchers’ networks. In this regard, future research should focus on gathering a more representative sample of HCP to render a national perspective on the COVID-19 burden on the Romanian healthcare system. Additionally, longitudinal designs could be implemented to allow comparisons between the experienced stress in different timeframes of the pandemic.

Second, we did not investigate potential gender differences in our sample’s appraisal of stress and bio-ethical principles. Future research could address this limitation by including similar proportions of male and female HCP in the sample.

In addition, the period in which the survey was completed (i.e., July–September 2021) and the channel of distribution (i.e., social media) could have influenced participants’ responses in the sense of dismissing the seriousness of the questions. Additionally, this being a retrospective study, data might have been biased by the current interpretation of events.

The authenticity of participants’ responses could have been influenced by desirability or impression management (e.g., wanting to create a positive impression by declaring not having broken any bio-ethical principles or not having made decisions affecting others). From a cultural standpoint, a lack of knowledge regarding bio-ethical principles could have influenced responses in our sample, limiting a clear representation of what abiding or infringing these principles might imply. In this regard, training programs addressing all categories of Romanian HCP could be salutary, focusing on applying bio-ethical principles and handling challenging situations in the COVID-19 era. Courses on bio-ethics are delivered as part of academic medical curricula and could be supplemented with other forms of training.

## 5. Conclusions

The present study analyzed results from a survey on Romanian healthcare practitioners concerning personal experiences with COVID-19, wellbeing and stress-related symptoms, and the appraisal of bio-ethical principles respected or infringed during the pandemic in 2020.

Our results revealed several characteristics of our sample: a professional identity marked by making decisions affecting others (i.e., patients, colleagues), a tendency to limit autonomy, and a certain detachment from their personal experience with the disease. Overall, HCP with more experience (i.e., over 30 years) tended to make more decisions affecting others. In comparison to previous studies, our sample demonstrated moderate general wellbeing and low to moderate levels of distress. From a bio-ethical perspective, a discrepancy was found between reporting social infringement of justice, autonomy, and truth and mostly denying personal infringement of any principle. Nonetheless, beneficence and nonmaleficence were the hardest principles to infringe in our sample. In line with previous studies, our HCP reported bio-ethical challenges encountered worldwide, such as neglecting chronic patients, limiting family access to the hospital, managing overcrowded hospital sections, and ensuring the necessary personal protection while interacting with patients. In contrast to other studies stressing the importance of following protocols, our sample mentioned the “blind” protocol application as one of the most encountered problematic situations.

## Figures and Tables

**Table 1 ijerph-18-12749-t001:** Demographic characteristics of the total sample (*N* = 97).

Variables	*N*	Frequencies
Region		
Banat	79	81.4%
Transylvania	10	10.3%
Bucharest	2	2.1%
Dobrogea	2	2.1%
Oltenia	2	2.1%
Muntenia	1	1%
Moldova	1	1%
Occupation		
Doctor/Physician	66	68%
Psychologist/Therapist	6	6.2%
Nurse	6	6.2%
Manager	5	5.2%
Pharmaceutical industry	3	3.1%
Pharmacist	2	2.1%
Academic	2	2.1%
Administration	1	1%
Student	1	1%
Other	5	5.2%
Years of experience in the medical field		
0–5	22	22.7%
5–10	11	11.3%
10–20	22	22.7%
20–30	27	27.8%
Above 30	15	15.5%

**Table 2 ijerph-18-12749-t002:** Distribution of types of personal experience with COVID-19 and decisions affecting others.

Variable	*N*	Percent of Cases
Personal experience with COVID-19		
Attended patients with COVID-19	49	50.5%
One or more family members/close friends had the disease	40	41.2%
Read information about the virus, disease, and vaccine	36	37.1%
Had the disease—asymptomatic/mild symptoms	17	17.5%
Suffered the loss of a family member/friend due to COVID-19	16	16.5%
Had no direct experience	15	15.5%
Had the disease—moderate/severe symptoms	9	9.3%
Had other types of experience with COVID-19	4	4.1%
Need to make decisions affecting others due to the pandemic		
Decisions affecting family members	61	62.9%
Decisions affecting patients	52	53.6%
Decisions affecting colleagues	25	25.8%
Decisions affecting subordinates/staff members	25	25.8%
No decisions affecting others	9	5.2%

**Table 3 ijerph-18-12749-t003:** Differences between categories of professional experience regarding decisions affecting others.

Kruskal–Wallis Test Results	Decisions Affecting Family Members	Decisions Affecting Patients	Decisions Affecting Colleagues	Decisions Affecting Subordinates/Staff Members	No Decisions Affecting Others
χ2	4.252	15.651	10.844	6.630	7.533
df	4	4	4	4	4
Sig.	0.373	0.004	0.028	0.157	0.110

**Table 4 ijerph-18-12749-t004:** Correlations between the COVID-19 experience and making decisions affecting others.

COVID-19 Experience	Decisions Affecting Others				
	Family	Colleagues	Subordinates/Staff	Patients	No Decisions
Had the disease—asymptomatic/mild symptoms	0.130	0.038	0.162	0.157	−0.147
Had the disease—moderate/severe symptoms	0.172	−0.026	−0.026	−0.130	−0.102
One or more family members/close friends had the disease	0.253 *	0.225 *	0.129	0.065	−0.196
Suffered the loss of a family member/friend due to COVID-19	0.054	0.246 *	−0.071	0.135	0.049
Attended patients with COVID-19	−0.035	0.159	0.065	0.526 **	−0.323 **
Had no direct experience	−0.144	−0.122	−0.056	−0.288 **	0.355
Read information about the virus, disease, and vaccine	0.237 *	0.133	0.035	0.030	0.049
Had other types of experience with COVID-19	0.052	0.115	0.233 *	0.089	−0.066

* Correlation is significant at *p* < 0.05. ** Correlation is significant at *p* < 0.01.

**Table 5 ijerph-18-12749-t005:** Means and standard deviations for stress-related variables.

Variables	*N*	Minimum	Maximum	Mean
General wellbeing	97	1 (very poor)	5 (very high)	2.93
Irritability	97	0 (never)	4 (always)	1.89
Concern	97	1 (sometimes)	4 (always)	2.23
Relaxation difficulties	97	0 (never)	4 (always)	1.8
Anxiety, unrest	97	0 (never)	4 (always)	1.81
Fearful anticipation of a negative event	97	0 (never)	4 (always)	1.41

**Table 6 ijerph-18-12749-t006:** Frequencies of ratings of the general wellbeing and stress-related variables throughout 2020.

Variable		*N*		Percent	
Rating of general wellbeing in 2020					
Very low		16		16.5%	
Low		12		12.4%	
Moderate		40		41.2%	
High		21		21.6%	
Very high		8		8.2%	
Stress-related variables					
	Never	Seldom	Moderate	Often	Always
Irritability	3 (3.1%)	26 (26.88%)	49 (50.5%)	17 (17.5%)	2 (2.1%)
Concern	0	19 (19.6%)	42 (43.3%)	31 (32%)	5 (5.2%)
Relaxation difficulties	7 (7.2%)	35 (36.1%)	27 (27.8%)	26 (26.8%)	2 (2.1%)
Anxiety, unrest	10 (10.3%)	27 (27.8%)	36 (37.1%)	19 (19.6%)	5 (5.2%)
Anticipation of a negative event	17 (17.5%)	38 (39.2%)	30 (30.9%)	9 (9.3%)	3 (3.1%)

**Table 7 ijerph-18-12749-t007:** Results of linear regressions—predictors of the general wellbeing in 2020.

Predictors	B	Std. Error	Exp (β)	t	Sig.
Model 1—Stress-related variables					
Irritability	−0.477	0.160	−0.331	−2.978	0.004
Concern	−0.321	0.180	−0.229	−1.789	0.077
Relaxation difficulties	0.001	0.164	0.001	0.007	0.994
Anxiety, unrest	0.143	0.181	0.127	0.787	0.433
Anticipation of a negative event	−0.131	0.167	−0.112	−0.786	0.434
Model 2—Personal experience with COVID-19					
Attended patients with COVID-19	0.331	0.281	0.144	1.178	0.242
One or more family members/close friends had the disease	0.245	0.276	0.105	0.887	0.377
Read information about the virus, disease, and vaccine	−0.221	0.272	−0.093	−0.812	0.419
Had the disease—asymptomatic/mild symptoms	0.220	0.334	0.073	0.659	0.512
Suffered the loss of a family member/friend due to COVID-19	0.143	0.337	0.046	0.425	0.672
Had no direct experience	0.779	0.421	0.245	1.853	0.067
Had the disease—moderate/severe symptoms	−0.148	0.430	−0.037	−0.344	0.732
Had other types of experience with COVID-19	−0.259	0.654	−0.045	−0.396	0.693

**Table 8 ijerph-18-12749-t008:** Assessment of bio-ethical principles.

Principles Infringed During the Pandemic	On a Social Level		On a Personal Level	
	N	Percent of cases	N	Percent of cases
Beneficence	15	15.5%	8	8.2%
Non-maleficence	26	26.8%	4	4.1%
Autonomy	42	43.3%	23	23.7%
Justice	46	47.4%	15	15.5%
Truth	38	39.2%	9	9.3%
None of the above	14	14.4%	57	58.8%
Which principle would be the hardest to infringe?	N		Percent	
Beneficence	32		33%	
Non-maleficence	24		24.7%	
Truth	18		18.6%	
Autonomy	14		14.4%	
Justice	9		9.3%	

**Table 9 ijerph-18-12749-t009:** Differences between categories of medical experience regarding principles’ infringement.

Variables	χ^2^	df	Sig.
On a social level			
Beneficence	0.300	4	0.990
Non-maleficence	4.788	4	0.310
Autonomy	2.343	4	0.673
Justice	1.675	4	0.795
Truth	2.739	4	0.602
None of the above	5.954	4	0.203
On a personal level			
Beneficence	5.449	4	0.244
Non-maleficence	2.487	4	0.647
Autonomy	10.509	4	0.033
Justice	2.686	4	0.612
Truth	4.441	4	0.350
None of the above	1.777	4	0.777

**Table 10 ijerph-18-12749-t010:** Difference between categories of professional experience in bio-ethical situations encountered throughout 2020.

Situation	*N*	Percent of Cases	Professional Experience			Attending COVID-19 Patients		
			χ^2^	df	Sig.	U	Z	Sig.
Limiting access of patients’ families to the hospital	46	51.5%	3.871	4	0.424	751	−3.545	0.000
Neglecting medical assistance of patients with chronic illnesses	37	41.1%	9.166	4	0.057	870	−2.624	0.009
Choosing between self-protection and compliance with professional duties	30	33.3%	2.715	4	0.607	892.5	−2.555	0.011
“Blind” application of protocols	29	32.36%	3.192	4	0.526	1062	−1.037	0.300
Hospital discharge of COVID-positive patients upon request	27	30%	3.640	4	0.457	722	−4.220	0.000
Pollution with single-use materials	26	28.9%	3.030	4	0.553	1169.5	−0.061	0.951
Neglecting non-COVID-19 patients	23	25.6%	5.712	4	0.222	866.5	−3.031	0.002
Managing hospital sections at surge capacity	21	23.3%	9.401	4	0.052	914.5	−2.645	0.008
Lack of intensive care technical utilities	18	20%	1.404	4	0.844	841	−3.59	0.000
Triage of patients	17	18.9%	3.327	4	0.505	913.5	−2.876	0.004
Stigma associated with COVID-19	16	17.8%	4.303	4	0.367	986	−2.133	0.033
Restricting access of non-COVID-19 patients to medical services	15	16.7%	3.627	4	0.459	913	−3.03	0.002
Infringing patients’ rights	14	15.6%	4.147	4	0.387	1082.5	−1.108	0.268
Obtaining informed consent in the correct way	13	14.4%	2.290	4	0.683	1009.5	−2.036	0.042
Hospital admission without the patient’s consent	12	13.3%	2.437	4	0.656	1033.5	−1.803	0.071
Conflict between personal beliefs and the professional line of conduct	12	13.3%	8.164	4	0.086	1173	−0.038	0.970
Administering the mandatory treatment	9	10%	1.258	4	0.868	1154	−0.316	0.752
Discharging a patient with delirium upon request	7	7.8%	2.477	4	0.649	1150	−0.419	0.676
Refusal to treat COVID-19 patients	7	7.8%	2.024	4	0.731	1105	−1.143	0.253
Reluctance to express personal opinions as a healthcare worker	4	4.4%	2.386	4	0.665	1078	−2.053	0.040
No such situation	6	6.2%	2.295	4	0.682	1029	−2.542	0.011

## Data Availability

Data are available upon request.

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
