# Peer review of "Stress and Bio-Ethical Issues Perceived by Romanian Healthcare Practitioners in the COVID-19 Era"

_ijerph, 2021, doi:10.3390/ijerph182312749_

Round 1

Reviewer 1 Report

Methods must be better described especially regarding the study sample design, % of HCPs respondents with respect to the total of HCPs in Romania (also by region), % of non-respondents.

More information about the type of "social media" and the used methodology are needed.

Questionnaire and items have to be reported in Appendix.

Demographic characteristics of the sample are an important part of the results, not of the Appendix. 

Results shown in Table 4 and later are not informative if questions have not been asked in relation to the difference, for example, between the previous year and 2020 in the considered variables.

Line 228: underline that 58.8% is on a personal level

Analysis could be performed into two groups: respondents that attended patients with COVID-19 and that not attended patients with COVID-19 (even because some types of personal experience are scarce).   

Author Response

Thank you for the positive appraisal of our work. Following your important comments, we made the necessary changes in the manuscript. Please find our responses to your comments below, marked with blue font.

Methods must be better described especially regarding the study sample design, % of HCPs respondents with respect to the total of HCPs in Romania (also by region), % of non-respondents.

Response:  In line with your comment, we extended the empirical background with findings of several studies on Romanian HCP. In addition, in line with your request, we added information on the total of HCP in Romania and the percentage of our respondents with respect to the population of Romanian HCP (see lines 163-167). Unfortunately, we do not possess information on the distribution of HCP by regions, nor on the percentage of respondents who received the survey but did not complete it. While we acknowledge the reduced sample size as a limitation of our study (see lines 452-460), we highlight the importance of our results as they paint a different picture of some Romanian HCP attitudes to issues generated by the COVID-19 pandemic.

More information about the type of "social media" and the used methodology are needed.

Response: In line with your important specification, in the Materials and Methods section, we mentioned the site on which the survey was constructed and the types of social media used to distribute it (see lines 134-137). We also specified the inclusion criteria for HCP (line 143) and how the items were generated and extracted (lines 145-147).

Questionnaire and items have to be reported in Appendix.

Response: Thank you for your valuable specification. We appended the English translation of the questionnaire (see Appendix B).

Demographic characteristics of the sample are an important part of the results, not of the Appendix. 

Response: Thank you for your important comment. We inserted the demographic characteristics in the first section of Results (see line 169 and Table 1).

Results shown in Table 4 and later are not informative if questions have not been asked in relation to the difference, for example, between the previous year and 2020 in the considered variables.

Response: Thank you for your comment. We acknowledge that we did not investigate stress-related manifestations in HCP in the pre-pandemic period, nor do we possess data collected from that period that are representative of the general Romanian HCP population. Before 2020, the topic of stress-related symptoms in HCP has been largely overlooked in the Romanian research area. We found a sole study dating from 2013 investigating levels of stress in a sample of 100 Romanian nurses, reporting high levels of stress in 50% of respondents (Bidilica et al., 2013 - https://www.researchgate.net/publication/322887774). We added the longitudinal research of stress as a direction of research (see lines 458-460). Despite such limitations, we believe that our current data reflect the stress perceived by some HCP in a year that differed from the previous year so much that any comparison would be difficult. We believe that the perceived stress could partially be an expression of the general ethos or the period’s Zeitgeist, as the changes brought by the pandemic have disrupted the general population’s existential routine (e.g., vacation plans, mobility permissions, professional challenges, etc.).

Line 228: underline that 58.8% is on a personal level

Response: Thank you for your comment. We added the requested specification (line 255).

Analysis could be performed into two groups: respondents that attended patients with COVID-19 and that not attended patients with COVID-19 (even because some types of personal experience are scarce).   

Response: Thank you for your inspired suggestion. Following, we ran the analysis on bio-ethical challenges between HCP that attended and did not attend COVID-19 patients and obtained significant results on 13 situations (see lines 298-300). We inserted the results in table 10, and further interpreted them in Discussions (lines 428-430).

Reviewer 2 Report

Thank you for your interesting study. I only have one concern and that can be improved in the introduction section and has to be improved in the discussion/limitations: especially ethical attitudes have beneath SES, an age and gender dimension. As you do not have data on gender in your study, the minimum is to discuss the potential bias regarding previous studies.

Author Response

Thank you for the positive appraisal of our work. In line with your comment, we operated the necessary in the manuscript. Please find our response to your comment below, marked with blue font.

Thank you for your interesting study. I only have one concern and that can be improved in the introduction section and has to be improved in the discussion/limitations: especially ethical attitudes have beneath SES, an age and gender dimension. As you do not have data on gender in your study, the minimum is to discuss the potential bias regarding previous studies.

Response: In line with your specification, we added a paragraph in the Introduction summarizing previous findings on gender differences concerning ethical principles (see lines 99-104). We found little evidence of gender differences in prior work. However, we discussed our findings related to prior studies (see lines 418-421). We also added gender differences as a limitation of our study and a direction of future research (see lines 461-463).

Reviewer 3 Report

Stress and bio-ethical issues perceived by Romanian healthcare practitioners in the COVID-19 era

The COVID-19 pandemic will continue to be a research subject for quite a while still.  This paper is practically publishable as is.  All the necessary ingredients are present, and in state-of-the-art shape: introduction, study objectives, methodology, results, discussion, conclusion and references.

However, one would like to see some recommendations in line with the findings of this research.  For instance, the authors mention that “a certain level of ethical illiteracy is not to be discounted”.  Maybe some form of training on the-bio-ethical principles?

Despite that, this remains a publishable paper.

There are only a few minor mistakes which are indicated as comments in the PDF document attached.

This is a fine paper that deserves to be published.  One would like to see some recommendations in line with the results (see attached report).

Author Response

Thank you very much for the positive appraisal of our work. Following your specification, we added recommendations to the manuscript. Please find our response to your comment below, marked with blue font.

The COVID-19 pandemic will continue to be a research subject for quite a while still.  This paper is practically publishable as is.  All the necessary ingredients are present, and in state-of-the-art shape: introduction, study objectives, methodology, results, discussion, conclusion and references.

However, one would like to see some recommendations in line with the findings of this research.  For instance, the authors mention that “a certain level of ethical illiteracy is not to be discounted”.  Maybe some form of training on the-bio-ethical principles?

Despite that, this remains a publishable paper.

There are only a few minor mistakes which are indicated as comments in the PDF document attached.

This is a fine paper that deserves to be published.  One would like to see some recommendations in line with the results (see attached report).

Response: Thank you for pointing out the mistakes in the text. We operated the necessary corrections. In line with your valuable observation, we introduced directions for continuing our research and recommendations on bio-ethical training (see lines 473-477).

Reviewer 4 Report

The paper investigates the attitudes towards stress perceived by Romanian healthcare practitioners in the COVID-19 era. The paper provides some new information as it turns out that doctors in particular did not react very strongly to the onset of the pandemic. This is quite surprising. However, the paper suffers quite a bit from low external validity, owing to the sample. At least, the authors should do more in order to contrast the sample used in the paper to the entire spectrum of doctors and nurses in Romania. The authors are aware of these problems, but more information should be inluded in the paper. 

Author Response

Thank you for the appraisal of our work. In line with your comment, we operated changes in the manuscript. Please find our response to your comment below, marked with blue font.

The paper investigates the attitudes towards stress perceived by Romanian healthcare practitioners in the COVID-19 era. The paper provides some new information as it turns out that doctors in particular did not react very strongly to the onset of the pandemic. This is quite surprising. However, the paper suffers quite a bit from low external validity, owing to the sample. At least, the authors should do more in order to contrast the sample used in the paper to the entire spectrum of doctors and nurses in Romania. The authors are aware of these problems, but more information should be included in the paper. 

Response: Thank you for your observation. Following your comment, we extended the empirical background with findings of several studies on Romanian HCP. In addition, in line with your request, we added information on the total of HCP in Romania and the percentage of our respondents with respect to the population of Romanian HCP (see lines 163-167). Unfortunately, we do not possess information on the distribution of HCP by regions, nor on the percentage of respondents who received the survey but did not complete it. While we acknowledge the reduced sample size as a limitation of our study (see lines 452-460), we highlight the importance of our results as they paint a different picture of some Romanian HCP attitudes to issues generated by the COVID-19 pandemic.

Round 2

Reviewer 1 Report

Dear Authors,

the revised version of the mauscript can be accepted in the present form.

Kind regards